# FPGA-Based Processor for Continual Capacitive-Coupling Impedance Spectroscopy and Circuit Parameter Estimation

**DOI:** 10.3390/s22124406

**Published:** 2022-06-10

**Authors:** Akihiko Tsukahara, Tomiharu Yamaguchi, Yuho Tanaka, Akinori Ueno

**Affiliations:** 1School of Science and Engineering, Division of Electronic Engineering, Tokyo Denki, Saitama 350-0394, Japan; 2Department of Electrical and Electronic Engineering, Tokyo Denki University, Tokyo 120-8551, Japan; 21769@ms.dendai.ac.jp (T.Y.); yuho_tanaka@mail.dendai.ac.jp (Y.T.); ueno@mail.dendai.ac.jp (A.U.)

**Keywords:** electrical impedance spectroscopy, capacitive-coupling impedance spectroscopy, field-programmable gate array

## Abstract

In principle, the recently proposed capacitive-coupling impedance spectroscopy (CIS) has the capability to acquire frequency spectra of complex electrical impedance sequentially on a millisecond timescale. Even when the measured object with time-varying unknown resistance *R_x_* is capacitively coupled with the measurement electrodes with time-varying unknown capacitance *C_x_*, CIS can be measured. As a proof of concept, this study aimed to develop a prototype that implemented the novel algorithm of CIS and circuit parameter estimation to verify whether the frequency spectra and circuit parameters could be obtained in milliseconds and whether time-varying impedance could be measured. This study proposes a dedicated processor that was implemented as field-programmable gate arrays to perform CIS, estimate *R_x_* and *C_x_*, and their digital-to-analog conversions at a certain time, and to repeat them continually. The proposed processor executed the entire sequence in the order of milliseconds. Combined with a front-end nonsinusoidal oscillator and interfacing circuits, the processor estimated the fixed *R_x_* and fixed *C_x_* with reasonable accuracy. Additionally, the combined system with the processor succeeded in detecting a quick optical response in the resistance of the cadmium sulfide (CdS) photocell connected in series with a capacitor, and in reading out their resistance and capacitance independently as voltages in real-time.

## 1. Introduction

Electrical impedance measurement has attracted much attention in various fields. Electrical impedance spectroscopy (EIS) is a method of examining the amplitude (Ω) and phase (rad) of impedance (Ω) at each frequency (Hz). EIS can be applied to the medical and healthcare fields [1]. Furthermore, applications for monitoring substance properties include lithium-ion battery characteristic monitoring [2], aluminum alloy 7075 corrosion [3], and the dehydration of plants [4]. Particularly in the medical field, electrical impedance is applied in body tissue analysis as bioelectrical impedance analysis [5], disease diagnosis as bioelectrical impedance vector analysis [6], monitoring the contact status of local impedance in catheter ablation for treating arrhythmia [7], pathogen detection of COVID-19 [8], rapid screening for influenza virus [9], DNA analysis [10,11], continuous monitoring of bioimpedance [12], blood pressure monitoring [13,14], and more. Furthermore, electrical impedance tomography (EIT) [15] has been applied in monitoring lungs, brain activity, cancer diagnosis, and merging biomedical applications [16]. In the medical field, impedance measurement requires impedances to be obtained at multiple frequencies. Therefore, if impedance spectroscopy (IS) in a wide frequency band and in real-time is obtained, detailed information about living tissue can be obtained, indicating higher accuracy and faster illness diagnosis and treatment.

In conventional impedance measurement, the measurement object is connected to the measurement system via electrodes as resistant coupling, and the measurement frequency might be fixed. If the characteristics of the measurement object are known, then there is no problem; however, it is challenging to determine the appropriate frequency if they are unknown. Therefore, setting an appropriate frequency will be required for application.

Many studies have reported that electrical impedance can be measured using small devices, such as implant applications of bioimpedance [17] and wearable applications [18]. The components of the proposed devices comprise AD5933 [19] impedance converters by Analog Devices Inc., Norwood, MA, USA, microprocessors, field-programmable gate arrays (FPGAs) with reconfigurable large-scale integration, and application-specific integrated circuits (ASICs).

The AD5933 can measure impedance in 100 Ω to 10 MΩ with a maximum frequency of 100 kHz. A configuration using AD5933 and a microprocessor (ARM9S3C2440AL) were reported in [9] to measure impedance in 50 Ω to 1 MΩ at 100 Hz to 100 kHz for influenza virus detection. The relative error was less than 2% for a solid-state register at 100 Hz and less than 4.1% for influenza virus at 100 Hz. Ref. [11] reported that the AD5933 could be used for DNA and protein sensing by scanning 10 frequencies from 100 Hz to 100 kHz for 5.3 s per frequency sweep, with an accuracy of 0.02% or less. For devices based on AD5933, the maximum measurement frequency is limited to 100 kHz, and the measurement frequencies might be fixed. Therefore, a measurement system based on AD5933 might be challenging to apply in applications with high target frequencies or unknown characteristics.

A configuration using microprocessors has also been reported [20,21]. In [20], the configuration used a microprocessor (STM32F405RG, STMicroelectronics N.V.). The device could measure impedance from 1 Ω to 10 MΩ with 1% accuracy. The frequency was 1 mHz to 100 kHz using an analog-to-digital converter (ADC) and a digital-to-analog converter (DAC) in the STM32F405RG. The device in [21] could conduct measurement using an excitation voltage of 1 MHz with PSoC5 (Infineon Technologies AG, Cheonan-si, Korea).

The FPGA configuration is used when impedance at a high-frequency band is required. Ref. [22] reported EIS inductive measurement in the 10 MHz band, specifically for impedance cardiogram (ICG) and heart/respiration monitoring. The device could take 4000 measurements per second at a 100 kHz to 20 MHz frequency band using FPGA, high-performance ADC, and DAC. In [2], FPGA was used to synthesize the sine waves of 21 frequencies between 1 Hz and 1 kHz per second and apply them as an excitation voltage. Because the excitation voltage included frequencies of 21 points per second, the measurement time was short.

The ASIC configuration has been applied in detecting DNA hybridization [23] and measuring bioimpedance, which could be achieved with low power consumption [24]. Furthermore, ASIC configurations have been reported for EIT when impedance with multiple channels is required, such as prostate cancer screening [25], lung imaging [26,27], and many other purposes [28,29,30,31]. ASIC is effective when the measurement target is small, when there are many measurement points, and when low power consumption is desired. However, it is expensive for small-quantity production, and it is challenging to change the circuit.

Recently, we proposed capacitive-coupling impedance spectroscopy (CIS) [32] as a novel, noninvasive EIS method for an insulated resistor. CIS is known to have the capability to acquire frequency spectra of complex electrical impedance in a sequential manner on the millisecond timescale, even when the measured object with time-varying unknown resistance *R_x_*(*t*) is capacitively coupled with the measuring electrodes with time-varying unknown capacitance *C_x_*(*t*). In this method, the measurement object and the capacitive coupling are embedded in the nonsinusoidal oscillation circuit. The frequency spectra of the complex electrical impedance of the measurement object are obtained rapidly using discrete Fourier transform (DFT) to analyze the obtained oscillation waveform. Therefore, the resistance and capacitance of the measurement object can be estimated from the obtained impedance spectra, and changing the frequency of the excitation voltage is not required. Moreover, because the analysis is used in only a single-cycle waveform, EIS can quickly measure the measurement object, indicating that the time-varying unknown impedance is obtained with real-time processing. In [33], we reported good, estimated results using an equivalent circuit model comprising resistance and capacitance that could be applied in living tissue impedance measurement. We also reported the possibility of detecting rapid optical response in the resistance of cadmium sulfide (CdS) photocells.

In the conventional EIS method, it is often the case that sine waves with different frequencies are applied to the measurement object as excitation voltages, the currents flowing through the measurement object are measured, and the impedances for each frequency are obtained [1,3,4,11]. In this method, the excitation voltage needs to be switched multiple times and the measurement time increases in proportion to the number of excitation voltages. Since the excitation voltage needs to be applied for a single cycle or more, a certain measurement time is required as the frequency becomes lower. Furthermore, a method of synthesizing dozens of sine waves with different frequencies and applying a single excitation voltage has been proposed [2]. In this method, the frequency needs to be changed according to the measurement object. On the other hand, the CIS approach does not require the application of multiple waveforms with different frequencies, and EIS can be obtained by analyzing a nonsinusoidal oscillation waveform with a single cycle. There is no need to change the frequency according to the measurement object. Additionally, since the CIS approach is capacitive coupling, EIS can be measured in a non-contact manner through an insulator. An EIS method has been not reported that is based on capacitive coupling and does not require a frequency change of the excitation voltage.

In our previous studies [32,33], it was confirmed offline in [32,33] that the time-varying impedance can be obtained in milliseconds using the novel algorithm of the CIS approach. From this result, it was clarified that the CIS approach can obtain EIS on the order of milliseconds in principle. Therefore, the CIS method can be expected to be used for an electronic sensor that repeatedly measures an impedance spectrum via capacitive coupling in a short time. To embody the sensor, it is necessary to implement the CIS algorithm involving DFT in a digital processor and to determine circuit parameters, such as resistance and capacitance, from each impedance spectrum. Moreover, if these circuit parameters are output as an analog voltage, they can be handled simply. Thus, in this study, we aimed to develop a proof-of-concept prototype that obtains EIS based on the CIS approach, estimates circuit parameters, converts them to an analog voltage, and outputs them. In view of speeding up the above-mentioned processing, lowering the initial cost for modularization, and enhancing the function of circuit parameter estimation in the future, we adopted a FPGA-based processor for implementation. Furthermore, from the experiments reported in the previous paper [33], it was confirmed that the optical response of the CdS photocell occurs in a short time of about 2.5 ms. The frequency of the voltage excited by the CdS photocell was around 20 kHz. Thus, in our prototype, we aimed to support the frequency of 20 kHz or higher, and to obtain the result in 1 ms or less, including the CIS calculation, circuit parameter estimation, and analog voltage output as the execution time.

In this study, the target circuit model of the measurement object was a series circuit of capacitance and resistance (CR circuit model). The contributions of this study are as follows:The architecture of the proposed FPGA-based processor based on CIS and parameter estimation in the CR circuit model that can be measured in real-time were shown;The measurement accuracy of the proposed FPGA-based processor was confirmed;We show that continual impedance estimation in real-time is possible using the proposed FPGA-based processor.

Section 2 describes an overview of the proposed FPGA-based processor, the implemented CIS principle, the detailed configuration of the proposed processor, and the estimation of the processor execution time. Section 3 describes three experimental approaches conducted to evaluate the proposed processor, a comparison of the execution time with other environments, verification of the measurement accuracy using known fixed resistors and capacitors, and responsiveness verification to time-varying impedance by measuring a CdS photocell as an example. Section 4 presents the results of the three experiments, and Section 5 offers a discussion.

## 2. Materials and Systems

### 2.1. Overview of the Proposed FPGA-Based Processor and CIS Method

Figure 1 shows a block diagram of the constructed system composed of a nonsinusoidal oscillator involving a measured object *R_x_* in series with an unknown capacitance *C_x_*, a FPGA-based processor for CIS, and the parameter estimation on the CR circuit model. Table 1 shows the list of notations in this paper.

The non-sinusoidal oscillator circuit is based on a Schmitt trigger inverter (74HC14AP; Toshiba, Japan). The measurement object was targeted for resistance, and it was a model capacitively coupled with *C_x_*_1_ and *C_x_*_2_, where *C_x_* = *C_x_*_1_
*C_x_*_2_/(*C_x_*_1_ + *C_x_*_2_) denotes the combined capacitance of *C_x_*_1_ and *C_x_*_2_. The measurement object had a series circuit model of an unknown resistance *R_x_* and capacitance *C_x_* (CR circuit model). They were built in as part of the nonsinusoidal oscillator. The proposed FPGA-based processor comprised an input interfacing part, DFT, a spectrum calculation based on CIS, the parameter estimation on the CR circuit model, and a DAC. First, the oscillation waveform for a single cycle was digitalized and stored using the input interfacing part. Second, the impedance spectra were calculated using CIS calculations based on DFT. Third, the parameter estimation to fit the CR circuit model was obtained from multiple impedance spectra. Finally, the estimated results were converted to an analog voltage at a certain time, and the processor repeated these processes continually.

The measurement object was connected to a Schmitt trigger inverter oscillator via *C_x_*_2_ and to *R_A_* via *C_x_*_1_. *R_A_* existed for the current–voltage conversion. The unknown impedance ${\dot{Z}}_{x}$ could be calculated using Equation (1), and it could be estimated using the following process from [32,33]:(1)$${\dot{Z}}_{x}=\frac{{\dot{V}}_{12}}{{\dot{I}}_{12}}=\frac{{\dot{V}}_{12}}{{\dot{V}}_{1}/{\dot{Z}}_{A}}=\frac{{\dot{V}}_{12}}{{\dot{V}}_{1}}\cdot {\dot{Z}}_{A},$$
where ${\dot{V}}_{12}$ and ${\dot{I}}_{12}$ are the complex voltage and current of ${\dot{Z}}_{x}$, respectively, ${\dot{V}}_{1}$ is the complex voltage of ${\dot{Z}}_{A}$, and ${\dot{Z}}_{A}$ is the combined impedance of the known resistance *R_A_* and stray capacitance C_A_. ${\dot{Z}}_{x}$ at each frequency *f* can be obtained as follows:(2)$${\dot{Z}}_{x}\left(f\right)=\frac{{\dot{V}}_{12}\left(f\right)}{{\dot{V}}_{1}\left(f\right)}\cdot {\dot{Z}}_{A}\left(f\right).$$

In Figure 1, voltages $v_{1}\left(t\right)$ and $v_{12}\left(t\right)$ are nonsinusoidal oscillation waveforms. The Fourier series expansion of waveform for a single cycle ($T_{0}=1/f_{0}$) can be regarded as the sine wave of the fundamental frequency *f*_0_ and its integral multiple *nf*_0_ as follows:(3)$$v_{12}\left(t\right)=\overset{\infty }{\underset{n=-\infty }{\sum }}{\dot{V}}_{12}\left(nf_{0}\right)e^{j2\pi nf_{0}t},$$
(4)$$v_{1}\left(t\right)=\overset{\infty }{\underset{n=-\infty }{\sum }}{\dot{V}}_{1}\left(nf_{0}\right)e^{j2\pi nf_{0}t}.$$

Therefore, the Fourier coefficient of each frequency component can be obtained as the amplitude and phase spectra by the Fourier transformation of the section. Fourier transformations at continuous time *t* and frequency *nf*_0_ of $v_{1}\left(t\right)$ and $v_{12}\left(t\right)$ can be calculated as follows:(5)$${\dot{V}}_{12}\left(nf_{0}\right)=\frac{1}{T_{0}}\int_{0}^{T_{0}}v_{12}\left(t\right)e^{-j2\pi nf_{0}t}dt,$$
(6)$${\dot{V}}_{1}\left(nf_{0}\right)=\frac{1}{T_{0}}\int_{0}^{T_{0}}v_{1}\left(t\right)e^{-j2\pi nf_{0}t}dt,$$
where ${\dot{V}}_{12}\left(nf_{0}\right)$ and ${\dot{V}}_{1}\left(nf_{0}\right)$ are complex voltages at frequency *nf*_0_. If $v_{1}\left(t+T_{0}\right)=-v_{1}\left(t\right)$, $v_{12}\left(t\right)$ and $v_{1}\left(t\right)$ consist only of odd-order frequency components.

Because the proposed FPGA-based processor is a digital circuit, it was obtained using the DFT. The DFTs of ${\dot{V}}_{1}\left(nf_{0}\right)$ and ${\dot{V}}_{12}\left(nf_{0}\right)$ were calculated as follows:(7)$${\dot{V}}_{12}\left(nf_{0}\right)=\frac{1}{N}\overset{N-1}{\underset{i=0}{\sum }}v_{12}\left(i\right)e^{-j\frac{2\pi nf_{0}i}{N}},$$
(8)$${\dot{V}}_{1}\left(nf_{0}\right)=\frac{1}{N}\overset{N-1}{\underset{i=0}{\sum }}v_{1}\left(i\right)e^{-j\frac{2\pi nf_{0}i}{N}},$$
where $v_{1}\left(i\right)$ and $v_{12}\left(i\right)$ are the analog-to-digital (AD) converted values of $v_{1}\left(t\right)$ and $v_{12}\left(t\right)$, respectively, and *N* is the number of oscillation waveform samples at *f*_0_. The real and imaginary parts of ${\dot{V}}_{12}\left(nf_{0}\right)$ and ${\dot{V}}_{1}\left(nf_{0}\right)$ were calculated as follows:(9)Re{V˙12nf0}=1N∑i=0N−1v12icos2πnf0iN,
(10)Im{V˙12nf0}=−1N∑i=0N−1v12isin2πnf0iN,
(11)Re{V˙1nf0}=1N∑i=0N−1v1icos2πnf0iN, and
(12)Im{V˙1nf0}=−1N∑i=0N−1v1isin2πnf0iN,
respectively. Furthermore, the amplitude spectra ($\left|{\dot{V}}_{12}\left(nf_{0}\right)|\;\mathrm{and}\;|{\dot{V}}_{1}\left(nf_{0}\right)|\right)$ and phase spectra ($\theta_{V12}\left(nf_{0}\right)$ and $\theta_{V1}\left(nf_{0}\right)$) of ${\dot{V}}_{12}\left(nf_{0}\right)$ and ${\dot{V}}_{1}\left(nf_{0}\right)$ were respectively calculated as follows:(13)|V˙12nf0|=[Re{V˙12nf0}]2+[Im{V˙12nf0}]2,
(14)θV12nf0=−tan−1Im{V˙12nf0}Re{V˙12nf0},
(15)|V˙1nf0|=[Re{V˙1nf0}]2+[Im{V˙1nf0}]2,
(16)θV1nf0=−tan−1Im{V˙1nf0}Re{V˙1nf0},

Moreover, the amplitude spectra $\left|{\dot{Z}}_{A}\left(nf_{0}\right)\right|$ and phase spectra $\theta_{ZA}\left(nf_{0}\right)$ of ${\dot{Z}}_{A}\left(nf_{0}\right)$ were calculated as follows:(17)$$|{\dot{Z}}_{A}\left(nf_{0}\right)|=\frac{1}{\sqrt{{\left(\frac{1}{R_{A}}\right)}^{2}+{\left(2\pi nf_{0}C_{A}\right)}^{2}}},$$
(18)$$\theta_{ZA}\left(nf_{0}\right)=-\tan^{-1}\left(2\pi nf_{0}C_{A}R_{A}\right).$$

Therefore, unknown impedance ${\dot{Z}}_{x}$ was calculated using Equation (19) from the obtained amplitude and phase spectra:(19)$${\dot{Z}}_{x}\left(nf_{0}\right)=\frac{\left|{\dot{V}}_{12}\left(nf_{0}\right)\right|}{\left|{\dot{V}}_{1}\left(nf_{0}\right)\right|}\cdot |{\dot{Z}}_{A}\left(nf_{0}\right)|e^{j\left\{\theta_{V12}\left(nf_{0}\right)-\theta_{V1}\left(nf_{0}\right)+\theta_{ZA}\left(nf_{0}\right)\right\}}.$$

Then, the real and imaginary parts of ${\dot{Z}}_{x}\left(nf_{0}\right)$ were calculated as follows:(20)Re(Z˙x)=|V˙12nf0||V˙1nf0|·|Z˙Anf0|cosθV12nf0−θV1nf0+θZAnf0,
(21)Im(Z˙x)=|V˙12nf0||V˙1nf0|·Z˙Anf0sinθV12nf0−θV1nf0+θZAnf0.

When the measurement object was the CR circuit model, *R_x_* and *C_x_* were estimated by fitting an appropriate function to the real and imaginary parts of spectrum ${\dot{Z}}_{x}$ at the frequency *nf*_0_ (*n* = 1, 3, 5, …). The appropriate function for the real part was a constant (Equation (22)), and the function for the imaginary part was rational (Equation (23)).
(22)Re(Z˙x)=Rx,
(23)Im(Z˙x)=−12πnf0Cx.

Furthermore, the CIS method could be applied to bioimpedance by changing the circuit model of ${\dot{Z}}_{x}$ [33].

### 2.2. Architecture of the Proposed FPGA-Based Processor for CIS and the Parameter Estimation on the CR Circuit Model

The proposed FPGA-based processor comprised four blocks:(a)Input interfacing part;(b)CIS calculation;(c)Parameter estimation on the CR circuit model;(d)Digital-to-analog (DA) conversion.

All blocks were designed as digital circuits based on FPGA. However, part of blocks (a) and (d) comprised an external analog circuit. The processor operated as follows. In the (a) input interfacing part, the oscillation waveform was converted to input the voltage range of the ADC from the external analog circuit, and the analog voltage was converted by the ADC built into the FPGA. Then, the oscillation waveform for a single cycle was detected and stored in random-access memory (RAM). In the (b) CIS calculation block, the impedance spectrum was at the fundamental frequency *f*_0_ and odd order up to *M* based on DFT. *M* is a parameter that can be set from the external of the processor. In the (c) parameter estimation on the CR circuit model, the *R_x_* and *C_x_* of the measured object were estimated using the obtained impedance spectrum at each frequency based on the least-squares method. The estimations were performed from the obtained impedance spectrum data in four-cycle waveforms. In the (d) DA conversion block, the estimated *R_x_* and *C_x_* (*R_x_est_* and *C_x_est_*, respectively) were quantized and converted to pulse density modulation (PDM) signals based on 1-bit ΔΣDAC. Then, the CR low-pass filter (LPF) outputted the PDM signals of *R_x_est_* and *C_x_est_* as an analog voltage.

The digital circuits on FPGA were used as very-high-speed integrated circuits hardware description language (VHDL). The circuit design and simulation tool used was Vivado 2020.2 (Xilinx Inc., San Jose, CA, USA). The target FPGA was the Z-7020 (xc7z020clg400-1, Xilinx Inc.) of the Zynq-7000 series, and the FPGA board was Zybo Z7 (Digilent Inc., Pullman, WA, USA). The calculation module in the FPGA used single-precision floating-point arithmetic units for the four arithmetic operations. Furthermore, the trigonometric functions, angle adjustment parts, and polar transformations used to convert the amplitude and phase were CORDIC IP core (Xilinx Inc.). These were converted to a fixed-point format and calculated. The details of each block are described below.

#### 2.2.1. Input Interfacing Part

Figure 2 shows a detailed illustration of the input interfacing part. In the level converter block, voltage followers were connected to points $v_{1}\left(t\right)$ and $v_{2}\left(t\right)$ in the oscillator, and *C*_1_ removed the DC component. Then, these voltages were level-converted to approximately 1/2 using the inverted amplifier in the input voltage range of the ADC. The attenuation rate 1/2 of the inverted amplifier was experimentally set to fit within the input voltage range. *C*_1_ was 10 μF, *R*_1_ was 10 kΩ, and *R*_2_ was 5.1 kΩ. The op-amps were AD822 (Analog Devices Inc.). The power supplies of the circuit were 5 V from the mobile battery (CHE-061-IOT, TRA Co., Ltd., Osaka, Japan) and −5 V from the voltage converter IC (LTC1144, Analog Devices Inc.).$\;V_{\mathrm{ref}}$ was supplied at 0.5 V, half of the input voltage range of the ADC. Furthermore, the $v_{1}\left(t\right)$ and $v_{2}\left(t\right)$ circuits had the same configuration.

In the ADC and single-cycle waveform module detection, the ADC used the Xilinx ADC (XADC) built into the FPGA. The XADC is a successive approximation register type with a 12-bit resolution, 1 mega sample per second (sampling frequency *f_s_* = 1 MHz), input voltage range of 0–1 V, and differential input, and AD conversion is possible for two channels simultaneously. $v_{1}\left(t\right)$ and $v_{2}\left(t\right)$ after AD conversion were expressed as $v_{1}\left(i\right)$ and $v_{2}\left(i\right)$. The rising edge of the waveform from negative to positive was detected by monitoring $v_{1}\left(i\right)$. Moreover, this block had a frequency counter at the detected rising edge. Thus, the fundamental frequency *f*_0_ of oscillation waveform for a single cycle were calculated using the number of samples *N* during the rising edge and sampling frequency *f_s_* of the ADC (*f*_0_ = *f_s_*/*N*).

Simultaneously, when a rising edge was detected, the results of ADC were stored in the RAM until the next rising edge was detected. Furthermore, $v_{12}\left(i\right)$ was calculated by subtracting $v_{2}\left(i\right)$ from $v_{1}\left(i\right)$, and it was stored in RAM at the same time as $v_{1}\left(i\right)$. Moreover, two RAMs were prepared and processed as double buffering. When the first RAM completed storing waveforms for a single cycle, the CIS calculation block read the waveform data from the first RAM and started calculation. The next-cycle waveform was stored in the second RAM in parallel. Therefore, the storing of the waveform and the CIS calculations were performed in parallel.

#### 2.2.2. CIS Calculation

Figure 3 shows a detailed illustration of the CIS calculation module. The CIS calculation block comprised the DFT module for $v_{12}\left(i\right)$ and $v_{1}\left(i\right)$ and the calculated ${\dot{Z}}_{A}$ and ${\dot{Z}}_{x}$ modules. In these modules, the CORDIC IP core (Xilinx Inc.) was used for the trigonometric functions (sine and cosine) and the polar coordinate transformation (Equations (13)–(18)) to calculate the amplitude and phase spectra from the DFT results.

Because it is sufficient to focus on the odd-order spectrum and calculate the components required for estimation in the CIS calculation, calculating all frequency spectra is unnecessary. Thus, in the DFT definition (Equations (7) and (8)), this module calculated the real and imaginary parts (Equations (9)–(12)) in parallel by the required set *M*th-order. Furthermore, the trigonometric function results were shared because they could be commonly used in calculating $v_{12}\left(i\right)$ and $v_{1}\left(i\right)$. In the DFT calculation module, waveform data for a single cycle were read from the RAM and odd-order impedance spectra were calculated repeatedly.

${\dot{Z}}_{A}$ was calculated based on Equations (17) and (18). These calculations were polar transformations when the real part was $1/R_{A}$ and the imaginary part was $2\pi nf_{0}C_{A}$. Therefore, the module was configured to perform polar transformation using the CORDIC IP core. The *R_A_* and *C_A_* were measured in advance and set external to the processor as constants in the FPGA. ${\dot{Z}}_{x}$ was calculated based on Equations (20) and (21) using the results of the DFT and ${\dot{Z}}_{A}$, and the real part of ${\dot{Z}}_{x}$ (Re(Z˙x)) and the imaginary part of ${\dot{Z}}_{x}$ (Im(Z˙x)) at each frequency are output.

#### 2.2.3. Parameter Estimation on CR Circuit Model Based on the Least-Squares Method

In the proposed method, *R_x_* and *C_x_* were estimated by model fitting based on the least-squares method to the real and imaginary parts of the impedance spectra. To avoid the effect of noise, the model fitting used up to the *M*th harmonic component. The *R_x_est_* that minimized *J_Rx_* in Equation (24) was the estimated value of the *R_x_*, and the *C_x_est_* that minimized *J_Cx_* in Equation (25) was the estimated value of the *C_x_*.
(24)JRx=∑n=1M[Re{Z˙xnf0}−Rx_est]2, n=1,3,5,⋯,M,
(25)JCx=∑n=1MIm{Z˙xnf0}−−12πnf0Cx_est2, n=1,3,5,⋯,M.

In Equation (24), *J_Rx_* becomes the minimum when the derivative of *R_x_est_* becomes 0. In Equation (25), *J_Cx_* becomes the minimum when the derivative of *u* = 1/*C_x_est_* becomes 0.
(26)∂JRx∂Rx_est=∑n=1MRe{Z˙xnf0}−∑n=1M1Rx_est=0,
(27)∂JCx∂u=∑n=1M12πnf0Im{Z˙xnf0}+∑n=1M12πnf02u=0,

Therefore, estimated values can be obtained by solving Equations (28) and (29):(28)∑n=1M1Rx_est=∑n=1MRe{Z˙xnf0},
(29)∑n=1M12πnf02u=∑n=1M−12πnf0Im{Z˙xnf0}.

Let $Z_{xRe}$ and $Z_{xIm}$ be the real and imaginary part vectors of IS at *nf*_0_ (*n* = 1, 3, 5, …, *M*). Let $B_{Re}$ and $B_{Im}$ be the real and imaginary parts of the basis vector at *nf*_0_. These are defined as follows:(30)ZxRe=[Re{Z˙xf0}, Re{Z˙x3f0}, Re{Z˙x5f0},⋯, Re{Z˙xMf0}],
(31)ZxIm=[Im{Z˙xf0}, Im{Z˙x3f0}, Im{Z˙x5f0},⋯, Im{Z˙xMf0}],
(32)$$B_{Re}^{T}={\left(1,\;1\;\cdots ,\;1\right)}^{T},\;\mathrm{and}$$
(33)$$B_{Im}^{T}={\left(\frac{-1}{2\pi f_{0}},\;\frac{-1}{2\pi 3f_{0}},\;\cdots ,\frac{-1}{2\pi Mf_{0}}\right)}^{T},$$
respectively. Because the model fitting was performed to odd harmonic components, $B_{Re}$, $B_{Im}$, $Z_{xRe}$, and $Z_{xIm}$ were column vectors comprising *m* elements (*m* = *(M* + 1)/2). Equations (34) and (35) are shown below in a matrix form using $Z_{xRe},Z_{xIm},\;B_{Re}$**,** and $B_{Im}$.
(34)$$\left(B_{Re}^{T}B_{Re}\right)R_{x\_est}=B_{Re}^{T}Z_{xRe},$$
(35)$$\left(B_{Im}^{T}B_{Im}\right)u=B_{Im}^{T}Z_{xIm}.$$

From this matrix-form representation, the estimation values of *R_x_est_* and *C_x_est_* were calculated using the following equations,
(36)$$R_{x\_est}={\left(B_{Re}^{T}B_{Re}\right)}^{-1}B_{Re}^{T}Z_{xRe},$$
(37)$$C_{x\_est}=\frac{1}{{\left(B_{Im}^{T}B_{Im}\right)}^{-1}B_{Im}^{T}Z_{xIm}}.$$

The implementation of FPGA was simplified by substituting the values of the basis matrix into Equations (36) and (37), and they could be rewritten as follows:(38)$$R_{x\_est}=\frac{1}{m}\overset{m}{\underset{i=1}{\sum }}Z_{xRe\;i},\;\mathrm{and}$$
(39)$$C_{x\_est}=\frac{\sum_{i=1}^{m}B_{Im\;i}^{2}}{\sum_{i=1}^{m}B_{Im\;i}Z_{xIm\;i}},$$
respectively, where $Z_{xRe\;i}$ is the *i*th element of the vector $Z_{xRe}$. The same is true for $Z_{xIm\;i}$ and $B_{Im\;i}$.

Figure 4 shows a detailed illustration of the parameter estimation on the CR circuit model. This block calculates and updates *R_x_est_* and *C_x_est_* each time the block obtains a spectral value for each frequency *nf*_0_ (*n* = 1, 3, 5, …, *M*).

#### 2.2.4. DA Conversion

Figure 5 shows a detailed illustration of the DA conversion block. This block converted the estimated results of *R_x_est_* and *C_x_est_* into analog voltages and outputted them. First, the quantization module quantized the estimated results based on Equations (40) and (41). The quantization bits Q were 24 bits.
(40)$$V_{Rx}=R_{x\_est}\frac{2^{Q}-1}{R_{max}},$$
(41)$$V_{Cx}=C_{x\_est}\frac{2^{Q}-1}{C_{max}},$$
where *R_max_* is the maximum value of measurable resistance and *C_max_* is the maximum value of measurable capacitance. The output voltage was maximum when the estimated values were *R_max_* and *C_max_*. Moreover, the measurement range could be changed by changing the *R_max_* and *C_max_* values.

Second, the PDM signal converted the quantized values based on 1-bit ΔΣ modulation and outputted them to the external of the FPGA. This module primarily comprised a *Q* + 1 bit counter and register. The module added a quantized value and inverted MSB of the register, and MSB was outputted as the PDM signal. This module was operated at 10 MHz on the target FPGA board, where the ringing (unwanted voltage vibration) of the output waveform was slight.

Finally, the converted PDM signal was converted to an analog voltage using external CR LPF. The *R_LPF_* and *C_LPF_* were 10 kΩ and 3.3 nF, respectively. These values were adjusted so that half of the maximum output voltage of the FPGA board *V_max_* was outputted when $2^{Q-1}$, half of the maximum quantization value, was inputted. The conversion from the output voltage to the estimated values of *R_x_* and *C_x_* could be conducted using Equations (42) and (43):(42)$$R_{x\_est}=V_{Rx}\frac{R_{max}}{V_{max}},$$
(43)$$C_{x\_est}=V_{Cx}\frac{C_{\max}}{V_{max}}.$$

### 2.3. Execution Time of the Proposed FPGA-Based Processor

The proposed FPGA-based processor conducted calculated after acquiring the oscillation waveform for a single cycle. The number of clock cycles required to obtain *R_x_* and *C_x_* for one frequency spectrum was 449 + *N* cycles. In total, 449 + *N* cycles were the pipeline processing, and 449 cycles were pipeline stalls caused by waiting for the calculation result in the previous stage (data hazard). At the start of the DFT calculation, stalls occurred in 255 cycles, 119 cycles for ${\dot{Z}}_{x}$ sine and cosine calculation, and 75 cycles for parameter estimation part *C_x_est_* calculation. The total number of stalls was 449 (= 255 + 119 + 75) cycles. *N* was the number of samples of single-cycle waveforms. The execution time *T_exe_* of estimating *R_x_* and *C_x_* by odd order up to order *M* could be calculated using
(44)$$T_{exe}=\frac{M+1}{2}\left(449+N\right)T_{FPGA},$$
where *N* is the number of waveform samples and *T_FPGA_* is the reciprocal of the operating frequency of FPGA.

## 3. System Evaluation Method

The proposed FPGA-based processor was evaluated by comparing the execution time of the software program on a PC and a microprocessor. Accuracy verification was conducted by measuring metal film resistors and polypropylene film capacitors as fixed *R_x_* and fixed *C_x_*, and detecting the rapid optical response in resistance of a CdS photocell as time-varying impedance. In the following evaluation, we used the Z-7020 (xc7z020clg400-1, Xilinx Inc.) of the Zynq-7000 series as the target FPGA, and the FPGA board used Zybo Z7 (Digilent Inc.).

### 3.1. Calculation Method of the Execution Times and Other Environments for Comparison

First, the execution time of the proposed FPGA-based processor was compared with the execution time of a C program on a PC and a microprocessor. A laptop PC (32 × 22 × 2 cm) and the Raspberry Pi 4 Model B (Raspberry Pi foundation) microprocessor were used. Table 2 shows the specifications of the PC and Raspberry Pi. The C program was created as a single-thread program operating in the same processing as the proposed FPGA-based processor. The trigonometric functions and polar transformations in the C program were cosf, sinf, sqrtf, and atan2f as single-precision floating-point format functions. The C compilers were Visual Studio Professional 2019 (Microsoft Co., Washington, DC, USA) on the PC and GCC on the Raspberry Pi. The data were obtained by saving the oscillation waveform when the measurement targets were a metal film resistor *R_x_* = 1.0 kΩ and polypropylene film capacitor *C_x_* = 10 nF using an oscilloscope (TBS2104B, Tektronix Inc., Beaverton, OR, USA). The C program was estimated from the IS data to the 19th order (e.g., the CIS calculation was repeated 10 times) using these waveform data. The execution time of the proposed FPGA-based processor was calculated using the required cycles from Equation (44). Generally, the execution time of FPGA can be measured. However, because the execution time of the proposed processor varied with the number of samples of the input-single cycle oscillation waveforms depending on the resistance of the measured object, it was difficult to perform a precise comparison for this system. Thus, to match the conditions with the comparison environments, the FPGA execution time was calculated from waveform data with the same amount of data. The execution time of the proposed FPGA-based processor was compared with the averaged values of running the C program 10 times on the PC and Raspberry Pi. Because the C program ran on the operating system, the execution time varied depending on the program running in the background. Thus, we employed the mean value repeated 10 times as the execution time.

### 3.2. Measuring Methods of Fixed Resistors and Capacitors

Second, the CR circuit model as a measurement object was measured using the proposed FPGA-based processor. *R_x_* had metal film resistors of 0.2 k–10 kΩ, and *C_x_* had polypropylene film capacitors of 1, 5, and 10 nF. Furthermore, the orders were set to 31st order for *C_x_* = 10 nF and 19th order for *C_x_* = 5 nF. In the *C_x_* = 1 nF, the order did not exceed 100 kHz and was adjusted. The *C_A_* was 10 pF. Furthermore, *R_max_*, which determined the measurement range, was set to 1.0 kΩ when *R_x_* was 0.2 k–0.91 kΩ, *R_max_* was set to 10 kΩ when *R_x_* was 2.0 k–9.1 kΩ, and *R_max_* was set to 100 kΩ when *R_x_* was 10 kΩ. Similarly, *C_max_* was 10 nF when *C_x_* was 1 and 5 nF, and *C_max_* was set to 100 nF when *C_x_* was 10 nF.

The analog voltages of the estimated values were measured using an oscilloscope. The proposed FPGA-based processor was estimated using four-cycle waveforms (one segment). The measured voltages were converted to the resistance and capacitance values. These estimated values were compared with the values measured using a digital multimeter (GDM-8341, Good Will Instrument Co., Ltd., New Taipei, Taiwan).

### 3.3. Measuring Method of CdS Photocell for Time-Varying Impedance Measurement

Finally, an experiment similar to the impedance measurement experiment that changed with time shown in [33] was performed. The experiment involved detecting a rapid optical response in the resistance of a CdS photocell (Photocoupler LCR0203, Nantang Senba Optical and Electonic, Shenzhen, China) connected in series to a capacitor. Figure 6 shows the constant current circuit for the LED current control of the photocoupler. The CdS photocell was used as a photosensitive resistor, which was connected in series with unknown capacitance *C_x_* to the nonsinusoidal oscillator shown in Figure 1. An NJU7032D (Nissinbo Micro Devices Inc., Tokyo, Japan) was used as the operational amplifier. A 2N7000 (ON Semiconductor Co., Phoenix, AZ, USA) was used as a field-effect transistor. The combined capacitance *C_x_* = *C_x_*_1_
*C_x_*_2_/(*C_x_*_1_ + *C_x_*_2_) was used as the alternative capacitance of *C_x_*_1_ and *C_x_*_2_. *C_x_* was provided by a polypropylene film capacitor of 10 nF. We measured *R_x_* and *C_x_* when *R_max_* was set at 2 kΩ, *C_max_* was set at 100 nF, and *I_Ph_* = 0.4–10 mA by changing the *V_C_* = 0.08 to 2.0 V. Because the oscillation frequency changed near 20 kHz, the order was set to the 5th order because the frequency would be approximately 100 kHz. The *R_F_*, *R_A_*, and *C_A_* were 8.2 kΩ, 0.2 kΩ, and 0 pF, respectively. The proposed FPGA-based processor was estimated using four-cycle waveforms (one segment). The analog voltages of the estimated values were measured using the same measuring instrument described in Section 3.2. The measured voltages were converted to resistance and capacitance values.

## 4. Results

Table 3 demonstrates the usage of the FPGA resources of the proposed processor in the target FPGA (Z-7020). Z-7020 is a model with medium-sized FPGA circuit resources, and consumed up to 73% of the LUTs among those resources.

### 4.1. Result of the Comparison of Execution Times

Table 4 shows the comparison of execution times. The FPGA execution time was calculated using Equation (44). The operated FPGA frequency was 150 MHz (*T_FPGA_* = 6.67 ns). In the used waveform data for comparison obtained by the oscilloscope, the fundamental frequency *f*_0_ was 675 Hz (*T*_0_ = 1.48 ms), and the number of waveform samples *N* was 1852 at 1.25 MHz sample per second. The order *M* was 19th order. Therefore, the proposed FPGA-based processor could be calculated at 0.153 ms after saving the waveform for a single cycle using these results. On the other hand, the PC consumed 0.263 ± 0.029 ms (Min.–Max.: 0.217–0.310 ms) and took 1.71 ± 0.19 times (Min.–Max.: 1.41–2.02 times) longer than the proposed FPGA-based processor. From the time-saving value of 0.11 ms, which is the result in Table 4, it became obvious that the proposed FPGA-based processor could perform 0.11 ms faster on average than the recent CPU with a clock frequency difference of ≥10 times. Moreover, the Raspberry Pi consumed 2.551 ± 0.622 ms (Min.–Max.: 1.047–3.193 ms) and took 16.6 ± 4.06 times (Min.–Max.: 6.8–20.8 times) longer than the processor.

The period of oscillation waveform was 1.48 ms. Because the processing of the processor ended in a single cycle, it could output the result while acquiring the oscillation waveform. Theoretically, the lower the clock frequency, the less power is consumed. Therefore, the proposed FPGA-based processor could be expected to obtain the same response as a PC with less power than a PC.

### 4.2. Measurement Results of Fixed Resistors and Capacitors

Figure 7 shows an example of the input oscillation waveforms of $v_{1}\left(t\right)$ and $v_{2}\left(t\right)$, and resulting output voltages of *V_Rx_* and *V_Cx_* from the proposed FPGA-based processor when *R_x_* = 1.0 kΩ and *C_x_* = 10 nF. The circuit parameters of *R_x_* and *C_x_* were estimated and converted to the voltages for each segment (four cycles). Because the proposed FPGA-based processor was estimated using four-cycle oscillation waveforms, the voltage waveform *V_Rx_* was changed in each segment. *V_Rx_* and *V_Cx_* were the output voltage under the measurable maximum value of *R_max_* and *C_max_*, which were set at 10 kΩ and 100 nF, respectively. *V_Rx_* and *V_Cx_* were about 330 mV. By converting from Equations (42) and (43), the estimated *R_x_* and *C_x_* were 1.0 kΩ and 10 nF, respectively. From these results, it was confirmed that the proposed FPGA-based processor could estimate *R_x_* and *C_x_* values close to the true value, and output them.

Figure 8 shows the estimated results of the metal film resistors *R_x_* (0.2–10 kΩ) and polypropylene film capacitors *C_x_* (1–10 nF) using the proposed FPGA-based processor. The plots and error bars in Figure 8 are the mean values and standard deviations within 100 ms obtained using the oscilloscope (12.5 k samples per second). Each figure in Figure 8 is the (a) mean estimated *R_x_* value, (b) mean absolute error of the estimated *R_x_* value, (c) mean estimated *C_x_* value, or (d) relative error of the estimated *C_x_* value, respectively. From Figure 8b, the mean absolute errors of the estimated *R_x_* values were −150 Ω to 300 Ω. When *R_x_* was less than 9.1 kΩ, the errors were within ±150 Ω, and when *C_x_* was 5 and 10 nF, the errors were within ±100 Ω. When *R_x_* was 2.0 k to 9.1 kΩ, the errors were within 5%, and when *C_x_* was 1 nF, the errors increased to negative dozens of ohms. Furthermore, the standard deviations increased. However, the error was within ±150 Ω. When *R_x_* was 10 kΩ, the errors increased. Because the measurement range was set at a maximum of 100 kΩ, it was considered that the tendency was different from that of other *R_x_* values.

However, the mean relative errors of the estimated *C_x_* values were −6% to +1%, and when *C_x_* was 5 and 10 nF, the relative errors were ±1.5%. When *C_x_* was 1 nF, the relative errors increased to −3% or above; however, the maximum range was −6% or less. Moreover, for all *C_x_* values, the relative errors of *C_x_* increased as *R_x_* increased.

### 4.3. Measurement Results of CdS Photocell

Figure 9 shows an example of the measured result for a CdS photocell using the proposed FPGA-based processor when *I_Ph_* = 0.4 mA (Low) and *I_Ph_* = 10 mA (High). Figure 9a shows the input oscillation waveform of $v_{1}\left(t\right)$ and $v_{2}\left(t\right)$, resulting in output voltages of *V_Rx_* and *V_Cx_* from the proposed FPGA-based processor. Circuit parameters of *R_x_* and *C_x_* were estimated and converted to the voltages for each segment (four cycles). Figure 9b shows the enlarged view near 0 μs (−1000 μs to 1000 μs) in Figure 9a. In Figure 9a, the output voltage *V*_Rx_ changed according to the optical response in the resistance of the CdS photocell. Furthermore, in Figure 9b, the proposed FPGA-based processor outputted the estimated values at 200–300 μs. From these results, the processor could read out the time-varying resistance *R_x_* and the capacitance *C_x_* independently in real-time.

Figure 10 shows the resistance *R_x_* of each *I_Ph_* value for a CdS photocell during the high period using the proposed FPGA-based processor. Figure 10a shows the time–resistance curves of *R_x_* at each *I_Ph_* value, and Figure 10b shows the saturation value at each *I_Ph_* value. The circuit parameters of *R_x_* were estimated and converted to the voltages for each segment (four cycles). The estimated *R_x_* values according to the changed *I_Ph_* values were observed.

## 5. Discussion

In comparing the execution times (Table 4), the proposed FPGA-based processor had the shortest execution time. The PC and Raspberry Pi could also conduct processing sufficiently briefly, indicating that the CIS method could calculate IS in the order of milliseconds. Moreover, if the C program is improved to a multithread instead of a single thread, the execution time can become even shorter. However, an oscilloscope or a data-acquisition board will be needed to obtain the oscillation waveforms using a PC; hence, the system might become large. Similarly, external equipment or an analog input circuit are necessary to obtain oscillation waveforms when using a Raspberry Pi. The execution time of the proposed FPGA-based processor depends on the time of the oscillation waveform for a single cycle. In the processor, the calculation starts after the waveform for a single cycle is saved. The most time-consuming process is DFT, which was repeated (*M* + 1)/2 times in the current configuration. However, each frequency component can be calculated independently and parallel in CIS calculation. Therefore, the execution time can be reduced by preparing a CIS calculation block to parallelize the calculation using a FPGA with more circuit resources. These improvements are required to support higher frequencies, which is a future study. Furthermore, as a strict comparison of execution time, it will be necessary to perform a comparison including the process of converting to an analog voltage.

Figure 7 shows that the *V*_Rx_ and *V*_Cx_ oscillated ±5 mV, considered to be due to ΔΣDAC. By converting the oscillation of ±5 mV to resistance and capacitance, the resistance was ±1.52 Ω when *R_max_* was set at 1.0 kΩ, and the capacitance was ±0.0152 nF when *C_max_* was set at 10 nF. Therefore, these were errors of ±0.15%.

Figure 8 shows that, when *C_x_* was 5 and 10 nF, the mean absolute errors of the estimated *R_x_* value were ±5%, and the mean relative errors of the estimated *C_x_* value were ±1.5%. However, when *C_x_* was 1 nF, these errors increased due to the increasing oscillation frequency. The oscillation frequency *f*_0_ will be high with decreasing *C_x_*. When *C_x_* was 1 nF, *f*_0_ was in the order of several kHz. Above the 10th order, the oscillation frequency exceeded 100 kHz, and the relative errors increased. Therefore, the order was adjusted so that the oscillation frequency was less than 100 kHz. The sampling frequency of the ADC caused the error because of the change in errors related to the oscillation frequency. The sampling frequency of the ADC in the current processor was approximately 1 MHz. The harmonic components analyzed using DFT were approximately 500 kHz according to the sampling theorem. However, the upper-limit frequency required to ensure CIS calculation accuracy will reduce. Therefore, improvement can be expected when using a dedicated ADC IC with a high sampling frequency and high resolution, instead of a FPGA internal ADC. Furthermore, the oscillation frequency can be decreased by increasing the $R_{F}$. However, the current flowing through the measurement object would decrease with increasing resistance *R_F_* of a Schmitt trigger inverter. Therefore, the error should increase in the voltage waveform occurring in *R_A_*. The analog input circuit, sampling frequency, and ADC and DAC resolution affected the measurement accuracy of the proposed FPGA-based processor. In the DAC, improvement could be expected using a dedicated DAC IC or multibit ΔΣDAC instead of a simple 1-bit ΔΣDAC. These are future studies.

Figure 9 and Figure 10 show that, in the measurement of a CdS photocell, the time-varying impedance was estimated using the proposed FPGA-based processor. Furthermore, the resistance changes according to the changes in current *I_Ph_* were also measured. The estimation results were output from 200 to 300 μs. Therefore, there were 3000–5000 estimations per second. The results indicated that simultaneous measurements in real-time depended on the time-varying impedance changing with the analog voltage. Figure 9b shows that the output voltage was blunted at the transition of the estimated value because of the LPF effect of CR in the ΔΣDAC. In the future, if the oscillation frequency becomes high, the circuit constants of the resistors and capacitances that make up the LPF must be reviewed.

The estimated frequency was approximately 100 kHz in the proposed FPGA-based processor, and it was considered almost equivalent to AD5933. However, the proposed CIS method has the advantage of measuring resistance and capacitance simultaneously. In real time, although the execution time depends on the time of oscillation waveform for a single cycle, thousands of estimates in 1 s are possible, which is considered useful. However, we consider that improvements are required for the measurable impedance range, precision, and supported frequency range.

Even in our prototype, the resistance and capacitance values were obtained from IS with a frequency of hundreds of Hz to about 100 kHz in milliseconds order, and the size was within about 20 × 20 × 3. Although the size of the prototype was a bit large, for example, it could be expected to be applied in cancer screening [34] using EIS of 100–100,000 Hz to reduce the size of the measurement system. Furthermore, since the measurement time affected the frequency of the excitation voltage, it took 100 ms or more for measurement at a relatively low frequency, such as 100 Hz. Even in a prototype, it can be expected to obtain results in a shorter time than the conventional method of switching sine waveforms.

The prototype is the first approach, and the versatility of the prototype is not high compared to that of a PC. However, in the future, for example, we would like to examine application in portable multi-channel simultaneous measurement systems and sensor modules mounted on a mobile robot system. As the number of channels increases, the amount of calculation and the processing time increase. Devices mounted on a moving robot are required to be real-time and responsive. Even in such cases, it can be expected to obtain the same performance as a PC. Furthermore, we think that our prototype will need to be enhanced in consideration of the thresholds required for real-time system operation, and tuning will be necessary when using it in real applications.

The size of this system, including the proposed FPGA-based processor, was approximately 20 × 20 × 3 cm. Because this system was a prototype, a standard and simple FPGA evaluation board without advanced external ADC and DAC ICs was used. The FPGA evaluation board was the largest in our prototype system. If a dedicated board is developed, it will be much smaller and can be used in wearable devices. It is a future task to develop a dedicated small board for miniaturization that does not fundamentally disrupt the subject’s behavior in a wearable application.

The prototype can handle only one equivalent circuit. Thus, the applicable range may be narrowed. For this (lower) price, the customer will receive a narrowly specialized solution without the possibility of applying it in other areas of interest. However, an advantage of employing FPGA is that the circuit can be rewritten. The advantages of using FPGA will be demonstrated, and the usefulness of this study will be further enhanced if we increase the number of equivalent circuits that can be supported and provide a function to dynamically switch the equivalent circuit according to the measurement target. It is unnecessary to change the CIS calculation unit when changing the measurement object, and it can be handled only by changing the circuit parameter estimation unit. If these enhancements can be applied, the proposed FPGA-based processor can expect the realization of inexpensive and flexible equipment.

## 6. Conclusions

In the first approach, this study presents a dedicated processor that is implemented as a FPGA to perform CIS, estimation of *R_x_* and *C_x_*, and their digital-to-analog conversions at a certain time and to repeat them continually. Even if there is a change in the circuit model to be estimated in the future, it is possible to respond flexibly by reconfiguring the FPGA-based processor. By performing circuit parameter estimation based on IS, more detailed impedance changes for the measurement object can be monitored in real-time. Furthermore, the proposed FPGA-based processor was evaluated through three examinations. The proposed processor could execute 19th-order CIS in 0.153 ms and a whole sequence in the order of milliseconds. The execution time comparison with other environments showed that the execution time of the proposed FPGA-based processor was shorter than that of a PC and a microprocessor. Furthermore, the measurement results from metal film resistors and polypropylene film capacitors suggested that estimating the resistance *R_x_* and capacitance *C_x_* with reasonable accuracy is possible. The processor estimated fixed *R_x_* within a relative error of ±5% for the metal film resistor (2.0–9.1 kΩ) and fixed *C_x_*, and within ±6% for the polypropylene film capacitor (1–10 nF). Moreover, in the measurement of a CdS photocell, the proposed processor was detected to have a rapid optical response to the resistance of the CdS photocell. The results indicated that simultaneous measurements in real time depend on the time-varying impedance changing with the analog voltage. It is considered that the purpose of demonstrating the real-time measurability based on the novel algorithm CIS method has been almost achieved by using the prototype shown in this paper. As our prototype is a simple configuration, there is room for improvement in practicality and size.

Future studies will focus on improving measurement accuracy by improving the processor and further reducing the execution time by parallelizing the CIS block. Furthermore, we will examine application to different circuit models, and practicality and size improvements.

## Figures and Tables

**Figure 1 sensors-22-04406-f001:**
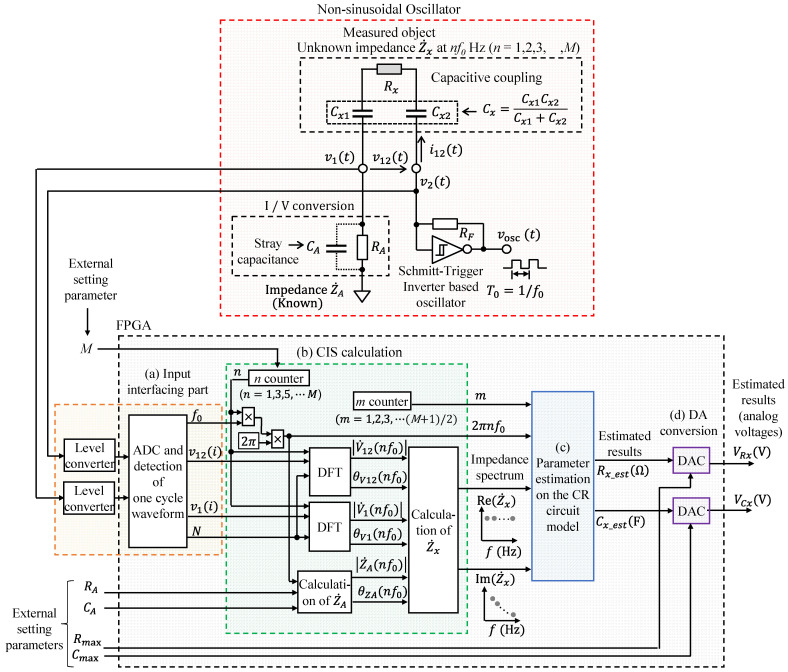
Block diagram of the constructed system composed of a nonsinusoidal oscillator involving the measured object *R_x_* in series with unknown capacitance *C_x_*_,_ a FPGA-based processor for CIS, and the parameter estimation on the CR circuit model.

**Figure 2 sensors-22-04406-f002:**
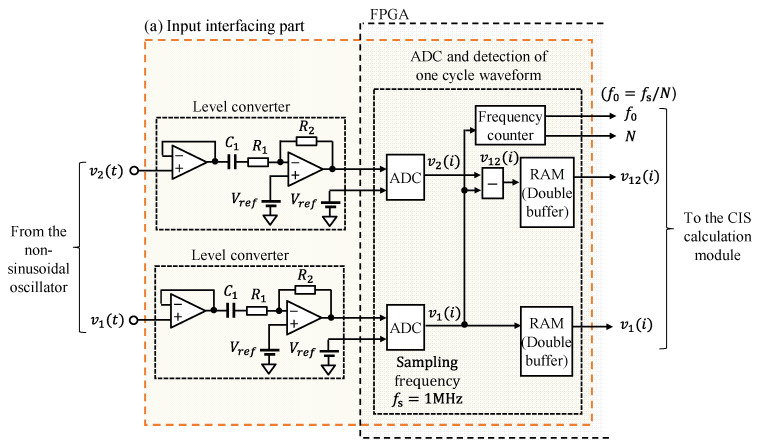
Detailed illustration of the input interfacing part written in Figure 1.

**Figure 3 sensors-22-04406-f003:**
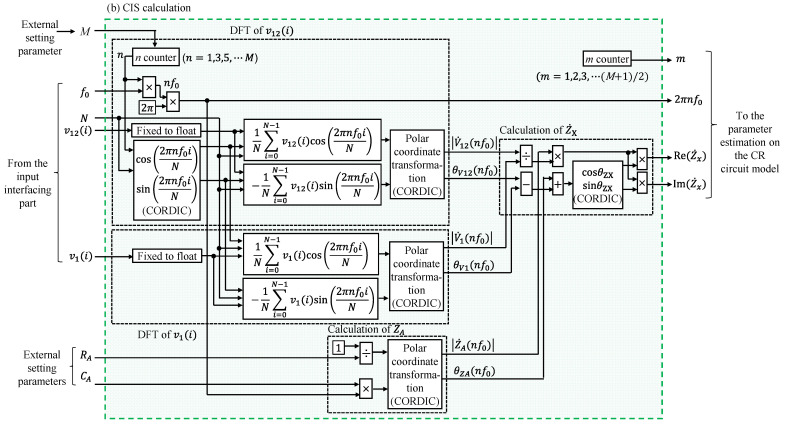
Detailed illustration of the CIS calculation module written in Figure 1.

**Figure 4 sensors-22-04406-f004:**
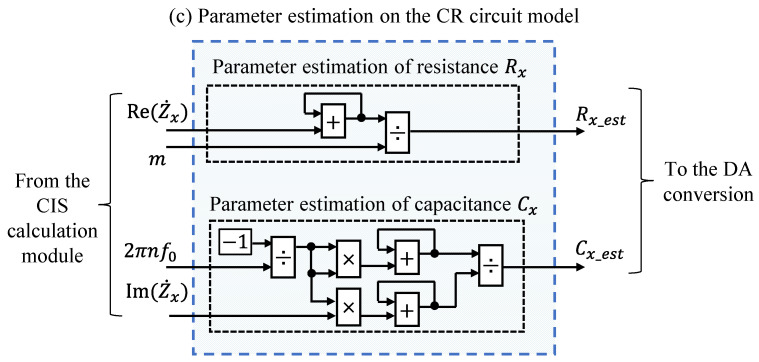
Detailed illustration of the parameter estimation on the CR circuit model written in Figure 1.

**Figure 5 sensors-22-04406-f005:**
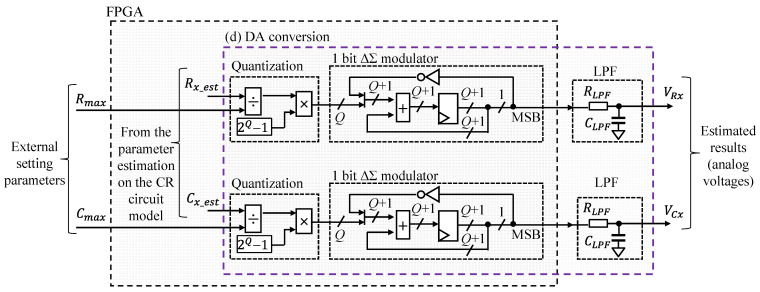
Detailed illustration of the DA conversion written in Figure 1.

**Figure 6 sensors-22-04406-f006:**
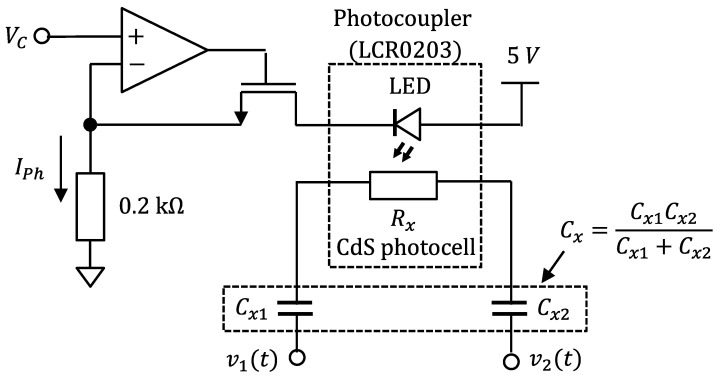
Constant current circuit for the LED current control of the photocoupler. The CdS photocell was used as a photosensitive resistor, which was connected in series with unknown capacitance *C_x_* to the nonsinusoidal oscillator shown in Figure 1.

**Figure 7 sensors-22-04406-f007:**
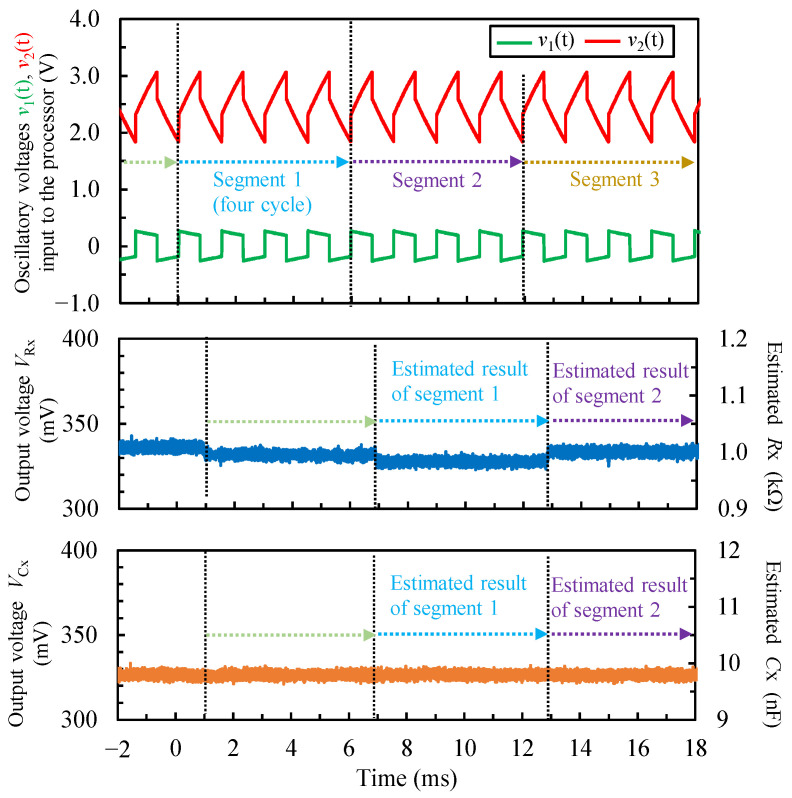
An example of oscillatory voltage $v_{1}\left(t\right)$ and $v_{2}\left(t\right)$ being inputted into the processor, resulting in output voltages of V_Rx_ and V_Cx_ from the proposed FPGA-based processor when R_x_ = 1.0 kΩ, C_x_ = 10 nF, M was set as 19th order, R_max_ was set at 10 kΩ, and C_max_ was set at 100 nF. The top plot is the output of the nonsinusoidal oscillator circuit when connected to the measurement object through capacitive coupling and the input waveform of the proposed processor. $v_{1}\left(t\right)$ and $v_{2}\left(t\right)$ have nothing else to convert. The bottom two plots show the results of the R_x_ and C_x_ circuit parameters estimated and converted into voltages for each segment (four cycles). These voltages were converted back into estimated values R_x_ and C_x_, and are shown on the right side of the y-axis as the estimated values corresponding to the voltage.

**Figure 8 sensors-22-04406-f008:**
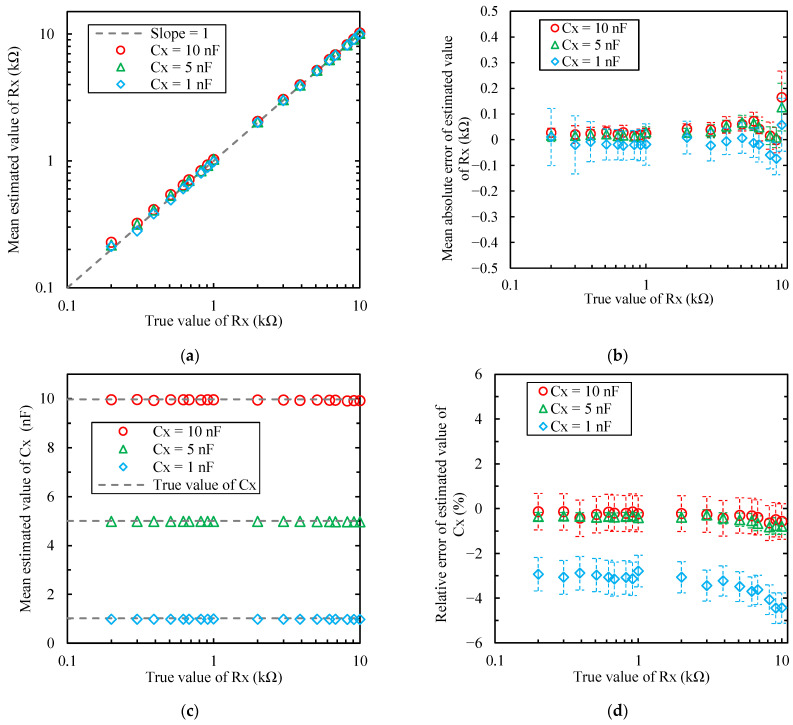
Estimated results of the metal film resistors *R_x_* (0.2–10 kΩ) and polypropylene film capacitors *C_x_* (1–10 nF) using the proposed FPGA-based processor: (**a**) mean estimated *R_x_* value; (**b**) mean absolute error of the estimated *R_x_* value; (**c**) mean estimated *C_x_* value; (**d**) relative error of the estimated *C_x_* value. Plots and error bars are the mean values and standard deviations within 100 ms obtained using the oscilloscope (12.5 k samples per second).

**Figure 9 sensors-22-04406-f009:**
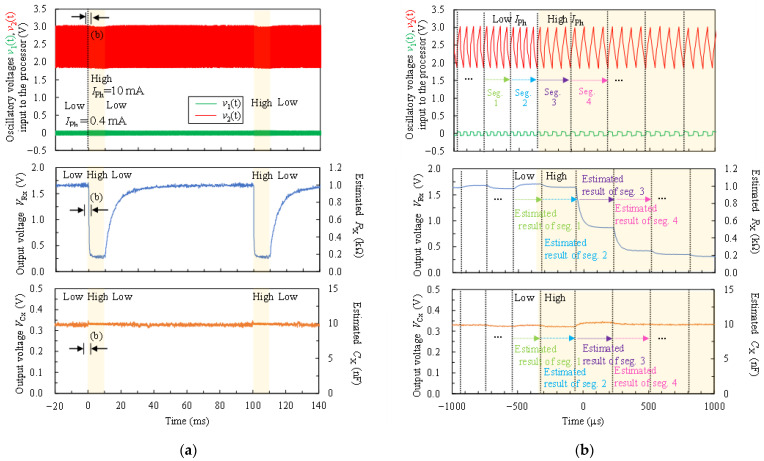
An example of the measurement result for a CdS photocell using the proposed FPGA-based processor when *I*_Ph_ = 0.4 mA (Low), *I*_Ph_ = 10 mA (High), *M* was set as 5th order, *R_max_* was set at 2.0 kΩ, and *C_max_* was set at 100 nF: (**a**) oscillatory voltage $v_{1}\left(t\right)$ and $v_{2}\left(t\right)$ input to the processor and resulting output voltages of *V_Rx_* and *V_Cx_* from the proposed FPGA-based processor. The top plot is the output of the nonsinusoidal oscillator circuit when connected to the measurement object through capacitive coupling and the input waveform of the proposed processor. $v_{1}\left(t\right)$ and $v_{2}\left(t\right)$ have nothing else to convert. The bottom two plots show the results of *R_x_* and *C_x_* circuit parameters estimated and converted to voltages for each segment (four cycles). These voltages were converted back into estimated values *R_x_* and *C_x_*, and are shown on the right side of the y-axis as the estimated values corresponding to the voltage; (**b**) enlarged view near 0 μs (−1000 μs to 1000 μs) in Figure 9a.

**Figure 10 sensors-22-04406-f010:**
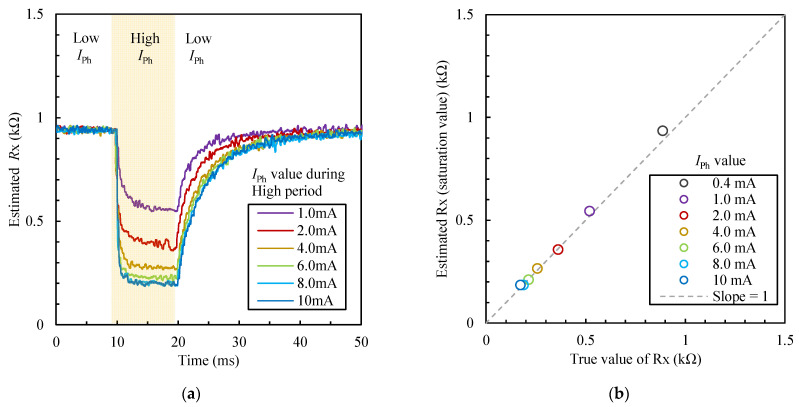
Resistance *R_x_* of each *I*_Ph_ value for a CdS photocell during the high period using the proposed FPGA-based processor: (**a**) time–resistance curves of *R_x_* at each *I*_Ph_ value; (**b**) saturation value at each *I*_Ph_ value. Circuit parameters of *R_x_* were estimated and converted to the voltages for each segment (four cycles).

**Table 1 sensors-22-04406-t001:** List of notations.

Symbol	Meaning
*R_x_*	Unknown resistance of the measurement object.
*C_x_* _1_	Unknown capacitance 1 of capacitive coupling to the measurement object.
*C_x_* _2_	Unknown capacitance 2 of capacitive coupling to the measurement object.
*C_x_*	Unknown capacitance caused by capacitive coupling. Combined capacitance of *C_x_*_1_ and *C_x_*_2_. *C_x_* = *C_x_*_1_ *C_x_*_2_/(*C_x_*_1_ + *C_x_*_2_).
${\dot{Z}}_{x}$	Unknown impedance of measurement object and capacitance caused by capacitive coupling. Combined impedance of *R_x_* and *C_x_*.
*R_A_*	Known resistance for I/V conversion. A resistance for calibration and converting the current flowing ${\dot{I}}_{12}$ through the measurement object into a voltage ${\dot{V}}_{1}$.
*C_A_*	Stray capacitance generated in parallel with *R_A_*.
${\dot{Z}}_{A}$	Known impedance. Combined impedance of *R_A_* and *C_A_*.
$v_{1}\left(t\right)$	Nonsinusoidal oscillation waveform generated by capacitive coupling to the measurement object. Voltage applied to ${\dot{Z}}_{A}$.
$v_{2}\left(t\right)$	Nonsinusoidal oscillation waveform generated by capacitive coupling to the measurement object.
$v_{12}\left(t\right)$	Excited voltage to the measurement object. $v_{12}\left(t\right)=v_{2}\left(t\right)-v_{1}\left(t\right)$.
$i_{12}\left(t\right)$	Current flowing through the measurement object.
${\dot{V}}_{1}$	Complex voltage of ${\dot{Z}}_{A}$ Complex number indication of $v_{1}\left(t\right)$.
${\dot{V}}_{12}$	Complex voltage of ${\dot{Z}}_{x}$ Complex number indication of $v_{12}\left(t\right)$.
${\dot{I}}_{12}$	Complex current of ${\dot{Z}}_{x}$ Complex number indication of $i_{12}\left(t\right)$.
*f*	Frequency.
*f* _0_	Fundamental frequency of nonsinusoidal oscillation waveform.
*T* _0_	Period of fundamental frequency *f*_0_. *T*_0_ = 1/*f*_0_.
*n*	Integer number.
${\dot{V}}_{1}\left(nf_{0}\right)$	Complex voltages ${\dot{V}}_{1}$ at frequency *nf*_0_.
${\dot{V}}_{12}\left(nf_{0}\right)$	Complex voltages ${\dot{V}}_{12}$ at frequency *nf*_0_.
Re{V˙1nf0}	Real part of ${\dot{V}}_{1}\left(nf_{0}\right)$.
Im{V˙1nf0}	Imaginary part of ${\dot{V}}_{1}\left(nf_{0}\right)$.
Re{V˙12nf0}	Real part of ${\dot{V}}_{12}\left(nf_{0}\right)$.
Im{V˙12nf0}	Imaginary part of ${\dot{V}}_{12}\left(nf_{0}\right)$.
$\left|{\dot{V}}_{1}\left(nf_{0}\right)\right|$	Amplitude spectrum of ${\dot{V}}_{1}\left(nf_{0}\right)$.
$\theta_{V1}\left(nf_{0}\right)$	Phase spectrum of ${\dot{V}}_{1}\left(nf_{0}\right)$.
$\left|{\dot{V}}_{12}\left(nf_{0}\right)\right|$	Amplitude spectrum of ${\dot{V}}_{12}\left(nf_{0}\right)$.
$\theta_{V12}\left(nf_{0}\right)$	Phase spectrum of ${\dot{V}}_{12}\left(nf_{0}\right)$.
$\left|{\dot{Z}}_{A}\left(nf_{0}\right)\right|$	Amplitude spectrum of ${\dot{Z}}_{A}\left(nf_{0}\right)$.
$\theta_{ZA}\left(nf_{0}\right)$	Phase spectrum of ${\dot{Z}}_{A}\left(nf_{0}\right)$.
Re(Z˙x)	Real part of ${\dot{Z}}_{x}$. Real part of the impedance of the measurement object.
Im(Z˙x)	Imaginary part of ${\dot{Z}}_{x}$. Imaginary part of the impedance of the measurement object.
*f_s_*	Sampling frequency of ADC.
*N*	Number of oscillation waveform samples at *f*_0_.
*M*	Upper limit order of fundamental frequency harmonics.
*R_x_est_*	Estimated *R_x_* by least-squares method.
*C_x_est_*	Estimated *C_x_* by least-squares method.
*J_Rx_*	Constant minimized by the least-squares method for *R_x_*.
*J_Cx_*	Constant minimized by the least-squares method for *C_x_*.
*u*	Constant with 1/*C_x_* replaced.
*m*	Number of vector elements. *m* = (*M* + 1)/2.
$Z_{xRe}$	Real part vectors of IS at *nf*_0_ (*n* = 1, 3, 5, …, *M*). One row *m* column vector.
$Z_{xIm}$	Imaginary part vectors of IS at *nf*_0_ (*n* = 1, 3, 5, …, *M*). One row *m* column vector.
$B_{Re}$	Real parts of the basis vector at *nf*_0_ (*n* = 1, 3, 5, …, *M*). *M* row one column vector.
$B_{Im}$	Imaginary parts of the basis vector at *nf*_0_ (*n* = 1, 3, 5, …, *M*). *m* row one column vector.
*V_Rx_*	Processor output voltage of *R_x_*. Estimated *R_x_* (*R_x_est_*) converted to voltage.
*V_Cx_*	Processor output voltage of *C_x_*. Estimated *C_x_* (*C_x_est_*) converted to voltage.
*Q*	Quantization bits of *R_x_est_* and *C_x_est_*.
*V_max_*	Maximum output voltage of the FPGA board.
*R_max_*	Maximum value of measurable resistance.
*C_max_*	Maximum value of measurable capacitance.
*R_LPF_*	Resistance of LPF used in the DA converter.
*C_LPF_*	Capacitance of LPF used in the DA converter.
$T_{exe}$	Execution time of the proposed FPGA-based processor.
$T_{FPGA}$	Reciprocal of operating frequency of FPGA.

**Table 2 sensors-22-04406-t002:** Specifications of the PC and microprocessor used for comparison of the execution time of CIS.

Item	PC	Raspberry Pi 4 Model B
CPU	Intel Core i7-1165G7@2.80 GHz, 1.69 GHz	Broadcom 2711 4-core ARM Cortex-A72@1.5 GHz
RAM	16 GB	8 GB
SSD	512 GB	–
OS	Windows 10 Home	Raspbian OS
C compiler	Visual Studio 2019	GCC

**Table 3 sensors-22-04406-t003:** Usage of the FPGA resources of the proposed processor in the target FPGA (Z-7020).

FPGA Resources	Available (Z-7020, xc7z020clg400-1)	Utilization
Look up table (LUT)	53,200	38,896 (73.1%)
LUT RAM	17,400	939 (5.4%)
Flip flop	106,400	56,011 (52.6%)
Block RAM	140	16.5 (11.8%)
DSP	220	72 (32.7%)

**Table 4 sensors-22-04406-t004:** Comparison of execution time for a single CIS when *R_x_* = 1.0 kΩ, *C_x_* = 10 nF, and *M* was set as 19th order *.

Environments		Execution Time (ms)	Ratio
FPGA-based proposed processor @150 MHz		0.153	1
PC (C, Single thread) @2.8 GHz, 1.69 GHz	Minimum	0.217	1.41
	Mean ± S.D.	0.263 ± 0.029	1.71
	Maximum	0.310	2.02
Raspberry Pi 4 (C, Single thread) @1.5 GHz	Minimum	1.047	6.8
	Mean ± S.D.	2.551 ± 0.622	16.6
	Maximum	3.193	20.8

* Values of the PC and the Raspberry Pi were the mean and standard deviation values repeated 10 times.

## Data Availability

The data that support the findings of this study are available from the corresponding author, A.T., upon reasonable request.

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
