# Peer review of "FPGA-Based Processor for Continual Capacitive-Coupling Impedance Spectroscopy and Circuit Parameter Estimation"

_sensors, 2022, doi:10.3390/s22124406_

Round 1
Reviewer 1 Report
The authors propose a capacitive-coupling impedance spectroscopy that is implemented on FPGA. The paper is generally well written and quite detailed. My specific comments for the authors are:
- Section 2.3 needs to be revised. Firstly, would be possible to describe the "449 cycles are the pipeline processing". In which modules are those cycles dedicated. Are there any cases of stalls?
- Also, how the M is calculated?
- In experimental results please describe the FPGA that were used and the resources that have been utilized. Please describe thoroughly the usage of LUT, DSP units etc.
- Section 3.1 states that is the "Comparison of the execution times". However, the comparison seems to be missing. The authors only report the PC and Raspberry resources.
- In all cases of the experimental results, which is the FPGA model that was used in the experimental results?
Reviewer 2 Report
Interesting topic and nicely processed article. As the article's main contribution, I see detailed block diagrams of the proposed solution and a precise mathematical apparatus used and presented by the authors.
Nevertheless, It would be nice if the authors' could complete the themes with answers to the following remarks/questions.
Lines 50 - 71 - Is it possible to briefly compare authors' published conclusions with the works of other authors? The work complements the previous publications of the authors, referred to [17] and [18]. The main disadvantage was the absence of real-time data processing (Lines 67 - 69). Based on lines 128-130, this task has been completed by the currently presented FPGA hardware design. What are the "practical" requirements for such hardware? Why is it not possible to solve this based on the performance of a standard personal computer/workstation? If the goal is to miniaturize wearable electronics, cannot HW with dimensions of 20x20x3 cm fundamentally disrupt the subject's behavior (and thus the measurement results)?
Why is FPGA execution time calculated? Is it possible to measure it? What are the expected differences between calculation and measurement?
What is the variability (statistical dispersion) of the times presented as mean values in Table 2? What does the time-saving 0.11 ms (0.263 - 0.153 ms) resulting from Table 2 mean?
What is the threshold (in ms) required for real-time system operation?
The price of the proposed solution is questionable. In the first approach, FPGA-based solutions will certainly be cheaper. Still, it should be mentioned that for this (lower) price, the customer will receive a narrowly specialized solution without the possibility of its application in other areas of interest (unlike PC, which use is universal).
Reviewer 3 Report
- The abstract seems dry, with a few math notations used. This does not encourage the readers to continue the paper. Please re-write it in a simpler way.
- The authors seemed to have missed the motivation of the proposed work in the first section. Please pay attention to this point.
- The authors need to add a paragraph at the bottom of Sec 1 to state the structure of the paper
- Although Fig1 and 2 are good high-level ones to read after maximising the page on the screen, but considering the hard copy of the paper, it is hard to spot the components. Please try to enlarge it or split it into smaller parts to make it easier for the reader to read.
- Over 40 equations exist in the paper, this is hard to grasp. I found it difficult to browse up and down to remember and link notations among those equations. Please add a table of notation and list all these notations in it
- Why the top plot of Fig 4 has no data on the right side of the y axis, compared to the other 2 plots underneath it?
- You certainly need to proofread the work
Round 2
Reviewer 1 Report
The authors have revised the manuscript and addressed sufficiently my comments.
Reviewer 3 Report
The authors have addressed the comments I raised previously. I have no other comments for this paper.